FPGA-based systolic deconvolution architecture for upsampling

Joseph Raj Alex Noel 1
Cai Lianhong 1
Li Wei 1
Zhuang Zhemin zmzhuang@stu.edu.cn 1
Tjahjadi Tardi 2
1 Department of Electronic Engineering, Shantou University , Shantou City , Guangdong Province , China
2 School of Engineering, University of Warwick , Coventry , United Kingdom
Hemanth Jude
Electronic publication date: 2022 May 11
Publication date: 2022
Volume: 8
Electronic Location ID: e973
Received 2021 Dec 29; Accepted 2022 Apr 14
Copyright: ©2022 Joseph Raj et al.
Copyright year: 2022
Copyright holder: Joseph Raj et al.
License: This is an open access article distributed under the terms of the Creative Commons Attribution License, which permits unrestricted use, distribution, reproduction and adaptation in any medium and for any purpose provided that it is properly attributed. For attribution, the original author(s), title, publication source (PeerJ Computer Science) and either DOI or URL of the article must be cited.
License URL: https://creativecommons.org/licenses/by/4.0/

Keywords: Upsample, Transposed convolution, FPGA, Deep learning

Funding: The Scientific Research Grant of Shantou University, China NTF17016 National Natural Science Foundation of China 82071992 Basic and Applied Basic Research Foundation of Guangdong Province 2020B1515120061 National Key R&D Program of China 2020YFC0122103 Guangdong Province University Priority Field (Artificial Intelligence) Project 2019KZDZX1013 This research was financially supported by The Scientific Research Grant of Shantou University, China, (Grant No: NTF17016); the National Natural Science Foundation of China (No.82071992); Basic and Applied Basic Research Foundation of Guangdong Province [grant number 2020B1515120061]; National Key R&D Program of China [grant number 2020YFC0122103] and the Guangdong Province University Priority Field (Artificial Intelligence) Project [grant number 2019KZDZX1013]. The funders had no role in study design, data collection and analysis, decision to publish, or preparation of the manuscript.

==============================
A deconvolution accelerator is proposed to upsample n × n input to 2n × 2n output by convolving with a k × k kernel. Its architecture avoids the need for insertion and padding of zeros and thus eliminates the redundant computations to achieve high resource efficiency with reduced number of multipliers and adders. The architecture is systolic and governed by a reference clock, enabling the sequential placement of the module to represent a pipelined decoder framework. The proposed accelerator is implemented on a Xilinx XC7Z020 platform, and achieves a performance of 3.641 giga operations per second (GOPS) with resource efficiency of 0.135 GOPS/DSP for upsampling 32 × 32 input to 256 × 256 output using a 3 × 3 kernel at 200 MHz. Furthermore, its high peak signal to noise ratio of almost 80 dB illustrates that the upsampled outputs of the bit truncated accelerator are comparable to IEEE double precision results.

Introduction

For the past decade, Deep Neural Networks (DNN) have been effectively employed in various applications of computer vision (Dongseok et al., 2019; Chen et al., 2014), speech recognition (Han et al., 2017) and image segmentation (Ronneberger, Fischer & Brox, 2015). Most of these applications concentrate on classification and segmentation problems. Convolutional layers form the primary modules of these DNN, where stacks of kernels are convolved with the input images to generate feature maps, that are subsequently passed through pooling and rectification layers to identify the dominant features (Ma et al., 2016). The process of convolution, rectification and pooling operations are repeated in a sequence till denser features are acquired from a larger receptive field. Finally, the feature maps are flattened and presented to a fully connected layer which provides a classification score (Zhang et al., 2015). Over the years researchers have attempted to implement a few notable DNNs on hardware, such as the AlexNet, VGG-16 (Lu et al., 2020) with lesser resources but higher throughput (Liu et al., 2018; Di et al., 2020; Lu et al., 2020). In general, these methods suffer from a common problem related to the usage of the pooling layer which gathers information from larger receptive field but loses the significant spatial coordinates from where the information has been obtained. To overcome this problem, DNN architectures incorporating encoder and decoder modules have been proposed, and amongst them U-Net proposed by Ronneberger, Fischer & Brox (2015) is the most popular model that is mainly used for segmentation applications. In the U-Net architecture, the feature maps that are downsampled in the encoder framework are later upsampled in the decoder stages. Furthermore, the decoder module of the U-Net and its variants include skip connections along with transpose convolution, also referred to as upsampler or deconvolution modules, to generate segmentation results of resolution equivalent to the input resolution (Ronneberger, Fischer & Brox, 2015).

Although many hardware implementations have been produced for encoder module (which is similar to VGG-16 architecture (Lu et al., 2020)), there are very few implementations of the decoder module, which involves the bottle-neck associated with the transpose convolution operation. One of the earliest deconvolution implementations on hardware was proposed by Zhang et al. (2017), where reverse looping and stride hole skipping mechanisms respectively ensure efficient deconvolution through the selection of input blocks based on output space and the removal of fractional addresses within the looping procedures. The deconvolution accelerator used C-based Vivado HLS libraries where loop unrolling and pipelining techniques were introduced to exhibit parallelism on a Zynq-7000 series FPGA. Dongseok et al. (2019) presented a lightweight CNN segmentation processor that includes: (i) dilation convolutions (insertion of virtual zeros within the kernel elements) for normal convolutions; (ii) transpose convolutions (insertion of virtual zeros within the feature maps) for enlargement of the feature maps; and (iii) the use of region of interest (ROI) based selection algorithm to enhance the throughput of the segmentation model. Dongseok et al. (2019) reported that their model when tested on a segementation application reduced the operational cost by 86.6% and increased the throughput (GOPS) by 6.7 times. Lu et al. (2020) introduced the Fast Winograd algorithm (FWA) to reduce the arithmetic complexity involved in the convolution operations and thereby improve the performance of CNN implementations on FPGA. The FWA exploits the structural similarity of the input feature maps and transforms the convolution operations into Element-Wise Multiplication Manipulation (EWMM), which reduces the number of multiplications and increases the required number of additions. Di et al. (2020) extended the use of FWA for transposed convolution implementations on FPGA, where the feature maps presented to the TransConv module were extended (by padding and introducing zeros in between the elements) and decomposed into four smaller subblocks. By applying FWA in parallel to these subblocks, the convolution output was obtained through element-wise multiplication of the input elements with the corresponding kernel coefficients. A performance improvement of 8.6 times was reported. However, the method was inefficient since FWA is suitable only for small kernels (Shi et al., 2019).

A reconfigurable generative network acceleration (GNA) with flexible bits widths for both inputs and kernels weights was proposed by Yazdanbakhsh et al. (2018). Inter and intra processing element (PE) processing and cross layer scheduling mechanisms are engaged to support the computations in the convolution, deconvolution and residual blocks. The inclusion of the dual convolution mapping method (where convolutions are associated with the outputs and deconvolutions are mapped to the inputs) efficiently balances the PE workload in convolution and deconvolution modules. It also improves the utilization performance of the PEs by 61% when compared to traditional methods. The GNA reported a 409.6 giga operations per second (GOPS) at 200 MHz with 142 mW power consumption. A convolution and deconvolution architecture capable of generating segmentations outputs close to real time was presented by Liu et al. (2018). The deconvolution module does not require addition of zeros between the input elements and produces upsampled outputs through a series of operations viz: (i) multiplication of single input pixel with the kernels; (ii) addition of overlapped outputs; and (iii) removal of outputs along the borders. An automatic hardware mapping framework based MATLAB and C scripts was employed to select the best design parameters which were then used to generate the synthesizable HDL code for implementation on the Xilinx Zynq board. A U-Net architecture was implemented and its performance was compared with GPU and CPU implementations. It achieved the best power and energy performance with speed being second only to the GPU implementation. Chang & Kang (2018) presented a massively parallelized deconvolution accelerator, referred as the TDC method, obtained by transforming the deconvolution operator into the four sparse convolutions. To avoid the overlapping summation problem, the height and width of the input images have to be determined to generate output blocks that do not overlap. Also the method has a load imbalance problem caused by the weights of the decomposed sparse convolution filters. Later in Chang, Kang & Kang (2020), the same authors optimized the TDC by rearranging filters which enabled DCNN accelerator to achieve better throughput. When implemented using C-based VIVADO HLS tool, the optimised TDC achieved 108 times greater throughput than the traditional DCNN.

We propose an FPGA-based scalable systolic deconvolution architecture (for different n × n input and k × k kernels) with reduced number of multipliers and adders, requiring no additional padding or insertion of zeros in between the inputs. Our contributions are as follows:

1. We present a Register Transfer level (RTL) based deconvolution architecture capable of upsampling n × n input to 2n × 2n output when convolved with a k × k kernel. The proposed module can be used as a standalone or readily connected to a pipeline to represent the decoder framework of the U-Net or the deconvolution CNN. We present upsampled outputs for intervals 32 × 32 to 64 × 64; 64 × 64 to 128 × 128 and 128 × 128 to 256 × 256 and compare the bit width truncated FPGA results with those of double precision MATLAB outputs.

2. The proposed architecture is systolic and governed by a single reference clock. After an initial latency, an upsampled element is obtained at every clock pulse which is then streamed to the next stage of the pipeline for further processing. A pipelined version capable of generating 256 × 256 output from 32 × 32 input using 3 × 3 kernel requires only 826.55 µs when operating at the frequency of 200 MHz.

3. The proposed architecture is coded using Verilog HDL and hence is void of any additional overheads associated in mapping CPU based algorithm directly to FPGAs. Also, the deconvolution architecture includes simple hardware structures such as the shift registers blocks, counters, comparators and FIFOs and thus can be extended to provide upsampled outputs by convolving with different kernel sizes. We also present the relevant equations to upsample n × n to 2n × 2n using 5 × 5 and 7 × 7 kernels. Further in ‘Hardware Implementation of the Upsampling Pipeline’ we present the hardware implementation of upsampling an random 32 ×32 matrix to 256 ×256 using 3 ×3 filters.

This paper is organized as follows. ‘Upsampling Techniques’ introduces the upsampling techniques used in deep networks. ‘Deconvolution Hardware Architecture’ presents the implementation of 4 × 4 to 8 × 8 deconvolution architecture. ‘Design of Experiments’ presents the experiments related to bit width requirements. ‘Analysis of the Deconvolution Accelerator’ discusses the required computation time, computation complexity and comparison results with other deconvolution architectures. ‘Hardware Implementation of the Upsampling Pipeline’ illustrates the implementation of the upsampling pipeline and finally ‘Conclusion’ summarizes our contributions.

Upsampling Techniques

The following are the upsampling methods used in deep networks: (i) Interpolation techniques (Lee & Yoon, 2010); (ii) Max unpooling (Shelhamer, Long & Darrell, 2016); and (iii) Transpose Convolution (Chang, Kang & Kang, 2020). Interpolation techniques could be either K-Nearest Neighbours, Bilinear or Bicubic interpolation and Bed of Nails. The first two interpolation methods introduce new samples either through direct copying or by a distance based weighted averaging of the neighbouring inputs. With Bed of Nails, upsampling is performed by inserting zeros in the positions other than the copied input elements. Max unpooling operator introduced in the decoder pipeline acts opposite to the max pooling operation of encoder framework. During the forward pass, at each max pooling operation, the positional indices of the maximum values are stored and later, during decoding, upsampling is performed by mapping the inputs at each stage to the corresponding coordinates, with the rest being filled with zeros. This technique is employed in SegNet (Badrinarayanan, Kendall & Cipolla, 2017), where coordinates of the maximum values of the feature maps obtained during the forward pass are used for the unpooling process during the decoding stages. The above techniques, though simple and efficient have a fixed relationship between input and output, and therefore are independent of the associated data. Hence they find less usage in deep networks where generalization through learning from inputs is a fundamental requirement.

In recent years, many deep learning architectures employ transposed convolution for deconvolution. Transpose convolution can be regarded as the process of obtaining the input dimensions of the initial feature map with no guarantee of recovery of the actual inputs since it is not an inverse to the convolution operation (Liu et al., 2018). Upsampling using transpose convolution can be achieved by: (i) sparse convolution matrix (SCM) (Liu et al., 2015); and (ii) fractionally strided convolutions (FSC) (Zhang et al., 2017; Liu et al., 2018; Yazdanbakhsh et al., 2018; Chang & Kang, 2018; Di et al., 2020). In SCM based upsampling, the 2D convolution process can be regarded as the multiplication of a SCM with an input image I. The convolution operation for an 8 × 8 input image with a 5 × 5 kernel, to give a 4 × 4 valid convolution output O are given by (1) SCM=k0,0k0,1...000k0,0...0000...00⋮⋮⋱⋮⋮00...k4,4000...k4,3k4,4,I=d1d2d3d4⋮d64,O=r1r2r3⋮r16

(2) SCM16×64×I64×1=O16×1.

SCM represents the spatial position of the kernels when slided across the image, where k(0,0), k(0,1), k(0,2) ...k(4,4) denote the kernel values at corresponding positions. I64×1 is the flattened input to enable matrix multiplication and O16×1 denote the flattened output after matrix multiplication which is finally reshaped to O4×4. The number of rows and columns of SCM depend on the number of input and output elements, respectively. Using the above relations, the backward pass which recovers the input resolution (4 × 4 to 8 × 8) is trivial by transposing SCM, i.e.,  SCM64×16T×O16×1=I64×1. SCM or SCMT, which contains the positional coordinates of the kernel, defines the forward or transpose convolution.

The traditional convolution process can also be employed to upsample an n × n input to 2n × 2n output by convolving with a k × k kernel (Kk×k). As the kernel is strided across the input, the convolution operator has to provide contributions associated only with elements present within the k × k window. Thus, to maintain the connectivity pattern and obtain interpolated outputs, it is convenient to introduce zeros in between the input elements before convolution. This procedure introduces fractional level convolution commonly referred as FSC.

To upsample an input image In×n, an intermediate extended image El×l is created by: (i) insertion of (s − 1) zeros in between the input elements; (ii) padding zeros (p) around the boundaries; and (iii) padding zeros (a) along the bottom and right edges of the input In×n. Table 1 summaries the description of all the parameters and Fig. 1 illustrates El×l, where a = (n + 2p − k) mod s and p=k−12. Next, El×l is convolved with the corresponding kernel Kk×k to obtain the upsampled output Om×m, i.e., (3) Om×m=El×l⨁Kk×k,

Table 1 Summary of the parameters in deconvolution.

Parameter	Description	
n	Resolution of the input image	
l	Resolution of the extended image	
m	Resolution of the output image, m = 2n	
s	Forward computation stride s = 2 (Dumoulin & Visin, 2016)	
k	Size of the kernel	
p	The number of zero padding	
a	The number of zeros added to the bottom	
	and right edges of the input	

Figure 1 The image El×l is obtained by inserting 0′s (white grids) in between the elements of the input image In×n.

Cyan and red grids denote the padding zero parameters a and p, respectively. The arrows denote the input and kernel values at corresponding positions.

where ⨁ denotes the valid convolution operation, l = (2 × n − 1) + a + 2p and m = 2n = s × (n − 1) + a + k − 2p. To upsample I2×2 using K3×3, p = 1, a = 1, l = 6 and m = 2n = 4, i.e., O4×4. Thus, FSC can be readily employed to upsample an n × n input to a 2n × 2n output. Both SCM and FSC when used for upsampling require introduction of zeros (either in SCM or in E) and Table 2 illustrates the number of zeros added for different upsampling intervals.

Table 2 The number of zeros added for different upsampling intervals.

Upsampling interval	2 × 2
→4 × 4	4 × 4
→8 × 8	8 × 8
→16 × 16	16 × 16
→32 × 32	32 × 32
→64 × 64	
ZSCM
{n2 ×  (m2 − k2) }	28	880	15808	259840	4185088	
ZFSC
{l2 − n2) }	32	84	260	900	3332	
Notes.

ZSCM and ZFSC represents the amount of added zero required.

Thus, when implemented on hardware the redundant operations (due to the zeros) consume large resources which generally lowers the performance of the hardware. However, when compared across different upsampling intervals the SCM requires exponential padding of zeros along the rows and columns, and thus, like many hardware implementations (Liu et al., 2018; Di et al., 2020; Chang, Kang & Kang, 2020) we use FSC technique to upsample the inputs. Though the proposed method like Liu et al. (2018); Chang, Kang & Kang (2020) employs four convolution patterns for upsampling, but efficiently decomposes the filters kernels into four simple, efficient and independent equations that avoid the need for redundant zeros required for FSC based upsampling.

Deconvolution Hardware Architecture

To upsample an n × n input to 2n × 2n output using FSC requires the dilation of the input as explained in the previous section. However, in practice for hardware implementations, inserting and padding zeros are not viable. Thus the proposed architecture consists of the following modules:

1. A shift register (SR) module used for temporary buffering of the streamed inputs. The input passes through a series of flipflops (FFs), FF1 to FFn, in a systolic manner governed by a common reference clock.

2. PEs are used to compute interpolated outputs by multiplying the inputs from the shift registers with the stored kernel coefficients.

3. A Data Control module (DCM) which consists of 2 control switches (CSW1 and CSW2) and 4 FIFOs arranged in parallel. CSW1 facilitates the temporary storage of PE outputs and CSW2 enables the systolic streaming of the upsampled results.

The length of the FIFOs and SR module depends on the kernel size and the upsampling intervals, i.e., 4 × 4 to 8 × 8 or 8 × 8 to 16 × 16, etc., and Table 3 illustrates the size requirements for different kernel and upsampling intervals.

As the input data progresses at a prescribed data rate into the SR module of the deconvolution accelerator, the PEs multiply the input data with the corresponding kernel coefficient. The control switches of the DCM then enable efficient storage, retrieval and streaming of the upsampled data.

Overview of 4 × 4 to 8 × 8 deconvolution architecture

To upsample a 4 × 4 input to a 8 × 8 output using FSC, a temporary extended image E of size 10 ×10 is created by inserting zeros between the input elements (shown as white grids in Fig. 1), padding around the boundaries (shown as red grids) and along the right and bottom edges (shown as cyan grids). As the 3 × 3 kernel slides across E, the output is computed from four computational patterns expressed in colours: pink, blue, yellow and green. For example, when the kernel is placed at the top left corner of E, the output O1 shown as the pink grids, the output image O8×8 is computed by multiplying the input d1 with central element k5 of the kernel, i.e., (4) O1=K5×d1.

Table 3 The requirements for different kernel sizes and upsampling intervals.

Upsampling
Interval	4 × 4 → 8 × 8	8 × 8 → 16 × 16	16 × 16 → 32 × 32	32 × 32 → 64 × 64	64 × 64 → 128 × 128	128 × 128 → 256 × 256	
Kernel
size (k)	3 × 3	5 × 5	7 × 7	3 × 3	5 × 5	7 × 7	3 × 3	5 × 5	7 × 7	3 × 3	5 × 5	7 × 7	3 × 3	5 × 5	7 × 7	3 × 3	5 × 5	7 × 7	
No. of FIFOs
and PEs	4	4	4	4	4	4	4	4	4	4	4	4	4	4	4	4	4	4	
Length
of FIFO	16	16	16	64	64	64	256	256	256	1024	1024	1024	4096	4096	4096	16384	16384	16384	
Size of the
Flipflops	5	10	15	9	18	27	17	34	51	33	66	99	65	130	195	129	258	387	
Zero
padding	1	2	3	1	2	3	1	2	3	1	2	3	1	2	3	1	2	3	

Likewise, progressing with a stride of 1 along the row followed by the column, the interpolated elements corresponding to the 8 × 8 output is obtained from the 4 × 4 input. For example, when the kernel is strided along the row and column, the blue and yellow grids of O8×8 give the interpolated output O2 and O3, i.e., (5) O2=K4×d1+K6×d2

(6) O3=K2×d1+K8×d5.

Similarly the green grid denoted by O4 computes the output (7) O4=K1×d1+K3×d2+K7×d5+K9×d6.

Figures 2A–2D illustrate the four computation patterns, where k1, k2, k3, …, k9 respectively correspond to the 3 × 3 kernel coefficients 1, 2, 3, …, 9, and d1, d2, d3, …, d16 respectively denote the 4 × 4 input 1, 2, 3, …, 16. Thus, by extending the 4 × 4 input and employing Eqs. (4) to (7) we can compute the required 8 × 8 upsampled outputs.1

The deconvolution architecture to upsample a 4 × 4 input to a 8 × 8 output by convolving with a 3 × 3 kernel is shown in Fig. 1 and according to Table 3, the architecture requires: (i) SR module of length 5 to allow buffering and enable computations to be performed in parallel; (ii) 4 PEs to compute Eqs. (4) to (7); (iii) 4 FIFOs each of length 16 are used to store the upsampled outputs; and (iv) a DCM comprising of multiplexers and 4 counters (count1, count2, count3, count4) for indexing the row and columns of the input and output, respectively.

Figure 2 (A–D) The four colours correspond to different computation patterns that correspond to the colours within O8×8 in Fig. 1.

The white grids denote 0’s.

The length of the SR module is based on the kernel size and the input resolution. In general the length of the SR module (NumSR) is given by NumSR=k−12×n+k−12. For I4×4 and K3×3, the length of SR module is 5. Furthermore, the length each of the FIFO is fixed as n × n. Since the input is 4 × 4, the FIFOs have a length of 16.

The PEs are hardware wired for a particular upsampling interval and kernel size, and execute in parallel to compute one of Eqs. (4) to (7). For example, PE1 receives input from SR1 and PE2 receives inputs from both SR1 and D0. The input and output connections of each PEs and their associated kernel coefficients are shown Fig. 3, where SR1, SR2, SR4 and SR5 are respectively the outputs of the flip flops FF1, FF2, FF4 and FF5 of the SR module.

Figure 3 The hardware implementation: (A) PE1, (B) PE2, (C) PE3 and (D) PE4 corresponding to Eqs. (6), (7), (8) and (9), respectively.

To explain the operation of module we use the same inputs and kernel coefficients as shown in Fig. 1, and the timing diagram of the generation of the outputs for the first 24 clock cycles is shown in Fig. 4. Once signal De is enabled, the deconvolution accelerator is active and the input data (signal D0 in the timing diagram) enters the SR module and propagates forward through FF1 to FF5 at the positive edge of the clock.

Figure 4 Timing diagram illustrating the upsampling process of 4 × 4 to 8 × 8 for T = t0 to T = t24.

At time T = t2, both PE1 and PE2 simultaneously receive their input from D0 and SR1, respectively, which are then multiplied with their corresponding kernel coefficients of the K3×3 to present the outputs, O1 and O2, respectively, i.e., (8) PE1:O1=SR1×k5

(9) PE2:O2=D0×k6+SR1×k4.

Subsequently as the input data advances, between clocks T = t3 and T = t6 and employing just PE1 and PE2, the upsampled elements of the first row (Row1) of O8×8 are computed. Due to zero padding at the rightmost boundary of the extended image, the last computation within PE2 requires just the multiplication of SR1 × k4. This is achieved by employing a counter (count2) to track the column indices and notify the multiplexer as shown in Fig. 3B. The architecture of PE1 and PE2 are shown in Figs. 3A and 3B, respectively.

To compute the upsampled elements of Row2 and Row3, along with PE1 and PE2, PE3 and PE4 operate in parallel. At clock T = t6, all the PEs simultaneously receive their input (D0, SR1, SR4 and SR5) from the SR module which then gets multiplied with the corresponding kernel coefficients and to simultaneously produce the respective outputs. Figures 3C and 3D illustrate the architecture of PE3 and PE4 where (10) PE3:O3=SR1×k8+SR5×k2

(11) PE4:O4=D0×k9+SR1×k7+SR4×k3+SR5×k1,

Here, O3 and O4 represent the outputs of PE3 and PE4, respectively. The availability of the input data at every clock cycle and the parallel execution of PEs enable the deconvolution accelerator to compute all 16 interpolated outputs of Row2 and Row3 of O8×8 within 4 clock cycles, i.e., between T = t7 and T = t10. As the input data proceeds into the deconvolution module the elements of Row4 to Row7 are computed in the similar fashion. Finally, to compute Row8 of O8×8, (row index is traced using count1) only PE3 and PE4 execute in parallel and using Eqs. (10) and (11) produces upsampled outputs O3 and O4. Again, to compensate for the zero padding at the bottom and right edges, multiplexers and additional controls are provided within PE3 and PE4 as shown in Figs. 3C and 3D.

Thus, at each clock instance, the PEs produce simultaneous outputs: O1, O2 by PE1 and PE2 for Row1; O1, O2 O3, O4 by PE1, PE2, PE3 and PE4 for Row2 to Row7; and O3, O4 by PE3 and PE4 for Row8 are temporarily stored in 4 separate FIFOs, FIFO1, FIFO2, FIFO3 and FIFO4 as shown in Fig. 5. The FIFOs write and read commands are synchronised with the input clock of the accelerator module and a series of controls generated by the DCM enables effective writing and streaming of the upsampled outputs from the FIFOs.

Figure 5 3 × 3 kernel deconvolution accelerator for 4 × 4 to 8 × 8.

DCM of 4 × 4 to 8 × 8 deconvolution architecture

The DCM is shown in Fig. 6 and consists of two control switches CSW1 and CSW2 that assist in the generation of FIFO write and read commands, enabling temporary storage and retrieval of the data. CSW1 and CSW2 are controlled by counters count1 and count3 which track the row indices of the input and the outputs, respectively. The FIFO write cycle is as follows:

Figure 6 DCM module architecture.

Fr1, Fr2, Fr3 and Fr4 are the read enable signals for FIFO1, FIFO2, FIFO3 and FIFO4, respectively. TC and LC are transfer signal and line control signals, respectively.

1. To store Row1 of O8×8: Initially count1 = 0, CSW1 = 0, PE1 and PE2 execute in parallel and their corresponding outputs stored in FIFO1 and FIFO2, respectively. Also, FIFO3 and FIFO4 are write disabled.

2. To store Row2 to Row7 of O8×8: (Beginning T = t7) count1 increments from 1 to 3, CSW1 = 1, PE1, PE2, PE3 and PE4 execute in parallel, and all the FIFOs are write enabled. PE3 and PE4 are connected to FIFO1 and FIFO2 where as PE1 and PE2 are linked to FIFO3 and FIFO4. The FIFO inputs are interchanged to enable easier read of the outputs during the read cycle.

3. Finally for Row8 of O8×8: count1 = 4, CSW1 = 1, only PE3 and PE4 execute in parallel and their outputs are connected to FIFO1 and FIFO2.

The read operation is managed by CSW2 and the Read signal is asserted after a delay of β clocks cycles and after De = 1 where β = θ + FIFOdelay.θ (refer to ‘Computation time of single Deconvolution Accelerator’) represents the delay before a valid sample is available at the output of PEs and normally FIFOdelay = 2 clock cycles. Thus, to upsample 4 × 4 to 8 × 8 using a 3 × 3 kernel we set β to 3 (θ = 2, for details refer to ‘Computation time of single Deconvolution Accelerator’). Once the Read is asserted, count3 and count4 respectively track the number of rows and columns of O8×8 and the data is read from the FIFOs using separate signals (Fr1, Fr2, Fr3 and Fr4) that are controlled by line control (LC) and transfer control signals (TF), respectively, as shown in Fig. 6. With LC = 1 or 0, and based on the rising edge of the TF, the data is read from the corresponding FIFO in an orderly manner, i.e., (12) Fr1=!TF&&LC.

(13) Fr2=TF&&LC.

(14) Fr3=!TF&&LC.

(15) Fr4=TF&&LC.

where ! and && denote the logical NOT and logical AND operations, respectively. The FIFO read cycle is as follows:

1. Initially read Row1 of O8×8: count3 = 0, LC = 1 and TF is toggled for every clock cycle. The generated read signals, Fr1 and Fr2, using Eqs. (12) and (13) control the read operations of FIFO1 and FIFO2, respectively.

2. To read Row2 to Row8 of O8×8: Starting at T = t13, count3 increments from 1 to 7, LC increments for each update of count3 and TF is toggled for every clock cycle as shown in Fig. 4. If LC is 0, using Eqs. (14) and (15) the computed results are read from FIFO3 and FIFO4. When LC is 1, FIFO1 and FIFO2 are enabled for reading. Note that count3 is controlled by the column counter count4 which increments for every 0 to 2n − 1.

The read cycle of the DCM introduces a delay (DCMdelay) of 3 clock cycles before the outputs are streamed in a systolic manner regulated by a reference clock. The proposed deconvolution architecture can be extended for various upsampling intervals by just extending the number of FFs within the SR module. The number of the PEs remain the same but their inputs differ. The PE equations for different upsampling internals for different kernel size are given in Table A1.

Design of Experiments

The proposed deconvolution accelerator was implemented on the Xilinx XC7Z020 FPGA using the Hardware Descriptive Language, Verilog. The behavioural and structural models were analyzed, simulated and synthesized using Xilinx VIVADO 2017.4.2 For experiments, we have chosen kernels of size 3 × 3, 5 × 5 and 7 × 7; image resolutions 32 × 32, 64 × 64 and 128 × 128 and clock frequencies 200 MHz.

Kernel bit width

At the positive edge of a clock signal, the deconvolution accelerator receives a stream of pixels 8-bit width which propagates through the shift register and PEs. The inputs are multiplied with the corresponding kernel coefficients with the results stored in FIFOs. For hardware implementations, fixed point is the natural choice of data representation due to simplicity and less usage of hardware resources. Thus, the floating point kernel coefficients are converted to fixed point by using a scaling factor of 2f and expressing the output as (f + 1)-bit within the FPGA. Here the optimum f is chosen by comparing the metrics such as Root Mean Square Error (RMSE) and the Peak Signal to Noise Ratio (PSNR) for different combinations of 2f with the corresponding IEEE double-precision output. Table 4 illustrates the results, where the kernel coefficients were selected from the distribution of range between −1 to +1 by invoking Keras tool (He et al., 2015). Initially, when f = 7, 8 and 9, the RMSE is high but with increase in the precision (bit width of the kernel), the PSNR improves and RMSE lowers, suggesting that fixed-point calculations are comparable to those of floating point operations. A scaling factor of 211 gives acceptable PSNR of 78.52 dB (Rao et al., 1990) with a low RMSE of 0.0303 and indicates that the fixed-point result is close to the IEEE double-precision . Increasing the bit width above 12 resulted in no significant improvement in PSNR and therefore the bit width of the kernels was set to 12-bit (f = 11 and 1 sign bit). Therefore a kernel value of (0.13250548)10 was first multiplied by 2048 (211) and its result (271.37122304)10 was rounded to (271)10. Its equivalent fixed-point representation in 11-bit along with 1 sign bit (000100001111)2 was used to represent the filter coefficient.

PEs output bit width

To illustrate that a deconvolution architecture produces upsampled outputs with considerable accuracy, we compare the upsampled outputs at different upsamping intervals (from 32 × 32 to 256 × 256) with those of the corresponding MATLAB outputs. For a realistic comparison, an image with a flat Power Spectral Density (PSD) (e.g., a white noise) was chosen as an input and the metrics, PSNR and RMSE, were used to evaluate the model. Based on the experimental results of the previous section, the input and kernel bit widths were set to 10-bit and 12-bit, respectively. The output the PEs were varied between 8 to 12-bit and the upsampled results of the deconvolution accelerator was compared with the corresponding MATLAB outputs. Table 5 shows the results and it can be inferred that 10-bit output is sufficient since the PSNR averages more than 58 dB across all upsampling intervals. Further increasing the bit widths resulted in no significant increase in the PSNR but resulted in considerable increase in hardware. Therefore, the choice of 10-bit upsampled outputs is reasonable. With the kernel and input width set to 12-bit and 8-bit, the accelerator produces upsampled outputs of 22 maximum bits (computation within the PEs include both multiplication and addition), and therefore the upsampled elements are left shifted 11 times and the 9 most significant bits (MSB) bits in addition to the sign bit are stored in the respective FIFOs. The shift operation compensates the earlier 211 multiplication of the kernel coefficients.

Table 4 Comparison of different kernel bit widths with IEEE double-precision output.

f	Data
width	PSNR	RMSE	Maximum data
length required (bits)	PEs length
word required (bits)	
7	8	50.64	0.7489	18	16 to 18	
8	9	50.13	0.7943	19	17 to 19	
9	10	60.84	0.2315	20	18 to 20	
10	11	67.64	0.1058	21	19 to 21	
11	12	78.52	0.0303	22	20 to 22	
12	13	76.15	0.0397	23	21 to 23	
13	14	80.70	0.0235	24	22 to 24	
Notes.

BitWidths used by the designed accelerator are in bold.

Table 5 Resource utilization and comparison of IEEE double-precision output with different PEs output bit widths.

The input and kernel bit widths are set to 10-bit and 12-bit.

layer	PE Output bitwidth	RMSE	PSNR	LUT	FilpFlop	LUTRAM	
	8	36.7596	16.8234	408	489	6	
32 × 32	9	4.9180	34.2950	459	498	8	
↓	10	0.2908	58.8579	484	511	8	
64 × 64	11	0.2908	58.8579	502	528	9	
	12	0.2908	58.8579	534	534	10	
	8	38.8182	16.3501	417	517	16	
64 × 64	9	5.6721	33.0559	469	532	18	
↓	10	0.2895	58.8976	503	540	20	
128 × 128	11	0.2895	58.8976	540	565	22	
	12	0.2895	58.8976	594	580	24	
	8	39.3273	16.2369	481	565	32	
128 × 128	9	5.8764	32.7486	530	581	36	
↓	10	0.2877	58.9532	570	596	40	
256 × 256	11	0.2877	58.9532	609	614	44	
	12	0.2877	58.9532	657	629	48	
Notes.

BitWidths used by the designed accelerator are in bold.

Comparison of upsampled results of different kernel sizes obtained from a trained U-Net models

We compare the outputs of the deconvolution accelerator with the MATLAB versions for various input sizes on kernel coefficients obtained from a trained U-Net model and natural images obtained from various datasets. First, we upsampled an random image of size 32 × 32 image to resolutions: 64 × 64, 128 × 128 and 256 × 256 using a 3 × 3 kernel with a maximum and minimum values of 0.7219356 and −0.64444816. The kernel coefficients obtained from the corresponding decoder frame work of the U-Net are stored in a register as 12-bit fixed point representation (as explained in ‘Kernel bit width’) and the upsampled results of the previous stage are provided as inputs to the current stage. Figure 7A illustrates the upsampled images at each stage of the pipeline (32 to 256). Tables 6 and 7 respectively show the corresponding performance scores and the resource usage. Furthermore, Table 8 reports resource usage for individual deconvolution units employing 3 × 3 kernels. Next, the camera man and the Natural images are examined with similar interpolation intervals. To illustrate that the proposed model can be extended for different kernel sizes, we also present upsampled results (Figs. 7B and 7C) obtained from 5 × 5 and 7 × 7 kernel sizes with maximum and minimum coefficient values of 0.78133786, −0.7012087, 0.5295713 and −0.46372077, respectively. The 10-bit deconvolution accelerator output is compared with the corresponding IEEE double-precision outputs using the metrics RMSE and PSNR. The outputs across different upsampling intervals show low RMSE and high PSNR of almost 80 dB which are comparatively better than the 40 dB of maximum PSNR reported by Chang, Kang & Kang (2020). Thus the 10-bit deconvolution accelerator indeed produces upsampled outputs comparable to MATLAB results.

Figure 7 Row 1 illustrates the upsampled outputs from MATLAB R2019a, and row 2 presents the results from the FPGA.

(A) Upsampled outputs using 3 × 3 kernel, (B) Upsampled outputs using 5 × 5 kernel and (C) Upsampled outputs using 7 × 7 kernel.

Table 6 Comparision of upsampled outputs at three different stages of the pipelined architecture.

Kernel	3 × 3	5 × 5	7 × 7	
Input	Face image	Camera man image	Lena image	
Layers	32 → 64	64 → 128	128 → 256	32 → 256	32 → 64	64 → 128	128 → 256	32 → 64	64 → 128	128 → 256	
PSNR	64.9057	63.2796	62.6464	65.3418	87.2817	77.2278	63.3388	64.6432	60.7390	58.0268	
RMSE	0.2905	0.3503	0.3768	0.2763	0.0221	0.00703	0.3479	0.2994	0.4693	0.6413	

Table 7 Resource usage for upsampling 32 × 32 to 256 × 256 using a 3 × 3 kernel.

Resource	Utilization	Total	Percentage (%)	
LUT	2383	53200	4.48	
Flipflop	2257	106400	2.12	
BRAM	43	140	30.71	
DSP	27	220	12.27	
IO	10	125	8.00	
BUFG	4	32	12.50	
LUTRAM	327	17400	1.88	

Table 8 Resource usage for different deconvolution model using a 3 × 3 kernel.

Resource\upsample model	32 to 64	64 to 128	128 to 256	
LUTs (Total:53200)	484	509	591	
Slice Registers (Total:106400)	517	579	606	
DSPs (Total:220)	9	9	9	
BRAM (Total:140)	2	4	16	
Power	0.004W	0.011W	0.011W	

Analysis of the deconvolution accelerator

Computation time of single Deconvolution Accelerator

The total computation time (Ttotal) required in terms of clock cycles for upsampling is given by (16) Ttotal=TCT+θ,

where TCT is the time required to obtain 2n × 2n samples from a n × n input, θ denotes the delay before a valid sample is available at the output of the PEs. TCT is obtained as follows:

1. To compute Row1 of the 2n × 2n, PE1 and PE2 execute in parallel n times.

2. To compute Row2n of the 2n × 2n, PE3 and PE4 execute in parallel n times.

3. To computes rows Row2 to Row2n−1 of the 2n × 2n, PE1, PE2, PE3 and PE4 operate in parallel as batches represented by N with each batch executing n times.

Therefore (17) TCT=2×n+N×n,

where n denotes the input size and N is given by (18) N=Row2n−Row1−12.

The denominator indicates that 2 rows of the 2n ×  2n output are computed when the all the PEs execute in parallel. The initial delay θ depends on k and is given by (19) θ=⌈k+14⌉+1.

⌈ ⌉ denotes the ceiling operation. Figure 8 illustrates Ttotal and Table 9 tabulates θ, TCT and Ttotal for different upsampling intervals and kernels. Thus, using the 3 × 3 kernel to upsample 4 × 4 to 8 × 8, (substitute k = 3 in Eq. (19)), the first effective result at the output of the PEs (PE1 and PE2) is obtained after a delay of two clock cyles, (i.e., θ =2). Subsequently PE1,PE2 execute 4 times in parallel to compute the samples of Row1. For Row2 to Row7, all the PEs independently execute 4 times in parallel but in 3 pipelined batches (N = 3 as computed using Eq. (18)). Finally, for Row8, PE3,PE4 again execute 4 times in parallel. Substituting the execution cycles of PEs required to compute each row of the output along with N in Eq. (17), the computation time TCT can be found. Thus, to upsample 4 × 4 to 8 × 8; TCT = 20 (i.e., 2 × 4 + 3 × 4 = 20) clock cycles, and Ttotal = 22 clock cycles. The upsampled outputs are temporarily stored in the FIFOs and after an initial delay of β +  DCMdelay clock cycles are read simultaneously by initiating the FIFO read signals as in Eqs. (12) to (15). The time-to-read (TR) the upsampled elements is 2n × 2n for an n × n input since the upsampled elements are streamed in a systolic manner (1 output per clock cycle) in reference to the common clock.

Figure 8 Visualize the various parameters (N, n, θ and TP) of Eqs. (17) and (20) on two stage pipleined deconvolution framework.

T and D denote the clock cycles and the upsampling intervals, respectively.

Table 9 θ, TCT and Ttotal for different kernel size.

Kernel size	Upsampling intervals	θ (cycles)	Tct (cycles)	Ttotal (cycles)	
	4 × 4 → 8 × 8	2	20	22	
3 × 3	32 × 32 → 64 × 64	2	1056	1058	
	64 × 64 → 128 × 128	2	4160	4162	
	128 × 128 → 256 × 256	2	16512	16514	
	4 × 4 → 8 × 8	3	20	23	
5 × 5	32 × 32 → 64 × 64	3	1056	1059	
	64 × 64 → 128 × 128	3	4160	4163	
	128 × 128 → 256 × 256	3	16512	16515	
	4 × 4 → 8 × 8	3	20	23	
7 × 7	32 × 32 → 64 × 64	3	1056	1059	
	64 × 64 → 128 × 128	3	4160	4163	
	128 × 128 → 256 × 256	3	16512	16515	

Computation time for the Pipelined architecture

The DCM allows separate read and write controls of the FIFOs and thus the upsampled elements of deconvolution accelerator can be readily streamed to the next stages: 2n × 2n to 4n × 4n, 4n × 4n to 8n × 8n and so on to represent a pipelined architecture that is similar to the decoder module of the U-Net. The computation time for the pipelined (TP) deconvolution framework is given by (20) TP=D×β+DCMdelay+TR,

where D denotes the number of upsampling intervals, TR (time-to-read) is TR = (2D×n)2) and DCMdelay = 3, and β is the delay before the read signal (Read) is asserted (refer to ‘DCM of 4 × 4 to 8 × 8 deconvolution architecture’). To upsample 32 × 32 to 256 × 256 using K5×5, TP is computed by substituting D = 3, β + DCMdelay = 8 (β = θ + FIFOdelay; refer to Table 9 for θ and ‘DCM of 4 × 4 to 8 × 8 deconvolution architecture’ for FIFOdelay and DCMdelay, and TR = 65536 cycles ((23 × 32)2) in Eq. (20)). Thus, Tp = 65560 clock cycles (3 × 8 + (23 × 32)2). Furthermore, for example, if a clock frequency of 50 MHz is considered, then the TP of the three-stage pipelined deconvolution module capable of upsampling 32 × 32 to 256 × 256 is 1310.84 µs (65542 × 0.02 μs), thus achieving a frame rate of 763 fps (frames per second). Figure 8 illustrates TP for a two stage pipelined deconvolution framework (n × n to 4n × 4n).

Comparison of computation complexity of the proposed architecture with other deconvolution architectures

The total number of operation (multiplications and additions) required to complete the upsampling process represents the computation complexity of the model. For the proposed architecture the number of multipliers OPmul and adders OPadd required to upsample n × n to 2n × 2n using k × k kernel are given by (21) OPmul=n×k−18k−12−14k−12.

(22) OPadd=n×k−18k−12−14k−12−2n2.

The total operations OPtotal is given by (23) OPtotal=2n×k−18k−12−14k−12−4n2.

Table 10 shows the OPmul, OPadd and OPtotal for various upsampling intervals and kernel sizes. When compared with existing architectures(refer to Table 10) where the total operations are computed using k2n2 + 2k(k − s)(n − s) + (k2 − s2)(n − 2)2 (for Liu et al. (2018)) and (2 × k2 − 1) × n2 for (Zhang et al. (2017) and Yan et al. (2018)), the proposed deconvolution architecture reduces the required operations by a maximum of 20%. We attribute this reduction to the pipelined structure of the architecture which executes either 2 or 4 PEs in parallel per clock cycle to produce the interpolated outputs. Also, at any clock instance, the maximum number of multiplier employed by the accelerator using a kernel of size k × k is k2, which relates to the parallel execution of all PEs in a batch for rows 2 to 2n − 1. Furthermore from Table 10, we observe a significant reduction in operations when 3 × 3 kernel size is used for up-sampling which directly contributes to resource utilization.

Table 10 Comparision of total operation of Liu et al. (2018), Zhang et al. (2017) and Yan et al. (2018) with our method for different kernel size and upsampling intervals.

k × k	n × n	Method	OPmul	OPadd	OPtotal	% saving	
		Zhang et al. (2017) and Yan et al. (2018)	9216	8192	17408	20%	
	32 × 32	Liu et al. (2018)	9216	7572	16788	17%	
		our	9025	4929	13954	–	
		Zhang et al. (2017) and Yan et al. (2018)	36864	32768	69632	19%	
3 × 3	64 × 64	Liu et al. (2018)	36864	31508	68372	17%	
		our	36481	20097	56578	–	
		Zhang et al. (2017) and Yan et al. (2018)	147456	131072	278528	18%	
	128 × 128	Liu et al. (2018)	147456	128532	275988	17%	
		our	146689	81153	227842	–	
		Zhang et al. (2017) and Yan et al. (2018)	25600	24576	50176	10%	
	32 × 32	Liu et al. (2018)	25600	22840	48440	7%	
		our	24649	20553	45202	–	
		Zhang et al. (2017) and Yan et al. (2018)	102400	98304	200704	8%	
5 × 5	64 × 64	Liu et al. (2018)	102400	94776	197176	8%	
		our	100489	84105	184594	–	
		Zhang et al. (2017) and Yan et al. (2018)	409600	393216	802816	7%	
	128 × 128	Liu et al. (2018)	409600	386104	795704	6%	
		our	405769	340233	746002	–	
		Zhang et al. (2017) and Yan et al. (2018)	50176	49152	99328	8%	
	32 × 32	Liu et al. (2018)	50176	45804	95980	5%	
		our	47524	43428	90952	–	
		Zhang et al. (2017) and Yan et al. (2018)	200704	196608	397312	6%	
7 × 7	64 × 64	Liu et al. (2018)	200704	189804	390508	4%	
		our	195364	178980	374344	–	
		Zhang et al. (2017) and Yan et al. (2018)	802816	786432	1589248	4%	
	128 × 128	Liu et al. (2018)	802816	772716	1575532	4%	
		our	792100	726564	1518664	–	

We also compare our proposed architecture with other deconvolution architectures in terms of (i) total operations, (ii) clock cycles required to complete an upsampling interval, (iii) hardware usage, (iv) GOPS and (v) resource efficiency (GOPS/DSP). To have favourable comparison across all architectures, we compare a single deconvolution module based on fixed point representation capable of upsampling a 128 × 128 input to 256 × 256 using a 5 × 5 kernel and Table 11 shows the results. Here GOPS, which denotes the processing performance of the model is computed using Di et al. (2020): (24) GOPS=OPtotalTtotal×1Freq,

Table 11 Comparison with other deconvolution architectures employing 3 × 3 kernel.

Work	Di et al. (2020)	Liu et al. (2018)	Zhang et al. (2017)	Chang, Kang & Kang (2020)	Proposed	Proposed	
Deconvolution
Layer	128 to 256	128 to 256	128 to 256	128 to 256	128 to 256	32 to 256	
Platform	ZCU102	XC7Z045	XC7Z020	XC7K410T	XC7Z020	XC7Z020	
Precision
(fixed point)	16 #	16 #	12 #	13 #	10 || 12	10 || 12	
OPtotal	294912	259350	294912	1318212	227842	298374	
Ttotal	20062	120909	31507	36864	16386	16390	
Input Filpflops	49794	16384	16384	–	258	457	
Output buffer	131072	147456	65536	–	65536	86016	
Total of
DSP usage	12 #	13 #	9 #	9 #	9	27	
GOPS
(Freq-MHz)*	2.94 #(200)	0.429 #(200)	0.936 #(100)	4.644 #(130)	2.781(200)	3.641(200)	
Resource
efficiency	
(GOPS/DSP)*	0.245 #	0.033 #	0.104 #	0.516 #	0.309	0.135	
Power(W)	0.07	0.03	–	0.032	0.011	0.026	
Maximum
Frequency	200	200	100	130	200	200 	
Notes.

* GOPS and GOPS/DSP are comnputed on single channel.

# Results obtained directly from the reference.

where Freq denotes the frequency. From Table 11, it is evident that the proposed architecture uses fewer operations and therefore less hardware resources to upsample. Furthermore, the proposed architecture produces the best resource efficiency of 0.309 GOPS/DSP at 200 MHz. The lowest clock cycles are required to upsample a 128 × 128 input to 256 × 256 across all considered architectures. We attribute the improvement to the hardware design which benefits in the reduction of operations and produces a maximum operations saving of 23% (by comparing the OPtotal of Di et al. (2020)) which directly relates to lower usage of the hardware resources. Furthermore, the proposed deconvolution accelerator achieves GOPS = 3.641 and GOPS/DSP = 0.135 for the pipelined architecture 32 × 32 to 256 × 256.

Extension of the proposed Deconvolution Accelerator

Although traditional U-Nets are based on 3 × 3 (Shvets & Iglovikov, 2018) kernels, few architectures either employ 5 × 5 (Chang, Kang & Kang, 2020) or 7 × 7 (Badrinarayanan, Kendall & Cipolla, 2017) in their encoder–decoder pipeline. Thus, to allow reusability of the architecture, we present in Table A1, equations for different upsampling intervals for 3 × 3, 5 × 5 and 7 × 7 kernels. The number of PEs are the same, but the length of the SR module and the FIFOs differ(refer to Table 3). Thus, by rewiring the inputs to the PEs, different upsampling intervals using different kernels sizes are obtained.

Hardware implementation of the upsampling pipeline

Figure 9A illustrates the upsampling pipleline where 32 × 32 random input is upsampled to 256 × 256 output using ZYNQ AX7020 FPGA board. Here to avoid computational overheads, the 8 bit 32 × 32 input was initialized in ROM, and systolically presented to the deconvolution accelerator pipeline as shown in Fig. 9B. The upsampling results for each layer (64 × 64 and 128 × 128) along with final 256 × 256 output is shown in the display screen (Fig. 9A). The complete upsampling pipeline required 131μs when executed at 50 MHz clock frequency. Here Xilinx IP cores, namely, Block ROM (https://docs.xilinx.com/v/u/Yy8V_830YccMjYlS44XWXQ) and RGB to DVI Video Encoder (employing HDMI interface) (https://www.xilinx.com/support/documentation/application_notes/xapp495_S6TMDS_Video_Interface.pdf) were used for initialization of the inputs and display of the upsampled outputs.

Figure 9 (A) (left) Hardware setup and (right) zoomed illurstation of 32 × 32, 64 × 64, 128 × 128 and 256 × 256. (B) The block diagram of the hardware setup illurstaing the BROM, pipelined upsampling accelerators and display module.

Conclusion

We present an FSC based systolic deconvolution architecture capable of upsampling n × n input to 2n × 2n output using a k × k kernel. The standalone (128 × 128 to 256 × 256) and the pipelined versions (32 × 32 to 256 × 256) implemented using 3 × 3 on a Xilinx XC7Z020 platform, achieved an overall performance and resource efficiency of 2.781 GOPS and 3.641 GOPS, 0.309 GOPS/DSP and 0.135 GOPS/DSP, respectively. When compared with other deconvolution architectures, the proposed architecture requires the least number of operations (with a saving of 23%) which results in lower usage of hardware. Furthermore, the high PNSR value demonstrates that the 10-bit upsampled results of deconvolution accelerator are comparable to IEEE double-precision outputs. In addition, the proposed architecture has a high scalability (the length of FIFOs and SR module change but number of PEs remain same) to suit different upsampling intervals.

10.7717/peerjcs.973/table-A1 Table A1 Appendix: Equations for extending the deconvolution accelerator different upsampling intervals (n × n to 2n × 2n based different kernel sizes.

PE number		
	Equations of 3 × 3 kernel upsampling architecture	
PE1	SR1 ×  K5	
PE2	D0 ×  K6 + SR1 ×  K5	
PE3	SR1 ×  K8 + SRn+1 ×  K2	
PE4	D0 ×  K9 + SR1 ×  K7 + SRn ×  K3 +	
	SRn+1 ×  K1	
	Equations of 5 ×  5 kernel upsampling architecture	
PE1	SR2n+2 ×  K1 + SR2n+1 ×  K3 + SR2n ×  K5 +	
	SRn+2 ×  K11 + SRn+1 ×  K13 + SRn ×  K15 +	
	SR2 ×  K21 + SR1 ×  K23 + D0 ×  K25	
PE2	SR2n+1 ×  K2 + SR2n ×  K4 + SRn+1 ×  K12 +	
	SRn ×  K14 + SR1 ×  K22 + D0 ×  K24	
PE3	SRn+2 ×  K6 + SRn+1 ×  K8 + SRn ×  K10 +	
	SR2 ×  K16 + SR1 ×  K18 + D0 ×  K20	
PE4	SRn+1 ×  K7 + SRn ×  K9 + SR1 ×  K17 +	
	D0 ×  K19	
	Equations of 7 ×  7 kernel upsampling architecture	
PE1	SR2n+2 ×  K9 + SR2n+1 ×  K11 + SR2n ×  K13 +	
	SRn+2 ×  K23 + SRn+1 ×  K25 + SRn ×  K27 +	
	SR2 ×  K37 + SR1 ×  K39 + D0 ×  K41	
PE2	SR2n+3 ×  K8 + SR2n+2 ×  K10 + SR2n+1 ×  K12 +	
	SR2n ×  K14 + SRn+3 ×  K22 + SRn+2 ×  K26 +	
	SRn+1 ×  K24 + SRn ×  K28 + SR3 ×  K36 +	
	SR2 ×  K38 + SR1 ×  K40 + D0 ×  K42	
PE3	SR3n+2 ×  K2 + SR3n+1 ×  K4 + SR3n ×  K6 +	
	SR2n+2 ×  K16 + SR2n+1 ×  K18 + SR2n ×  K20 +	
	SRn+2 ×  K30 + SRn+1 ×  K32 + SRn ×  K34 +	
	SR2 ×  K44 + SR1 ×  K46 + D0 ×  K48	
PE4	D0 ×  K49 + SR1 ×  K47 + SR2 ×  K45 +	
	SR3 ×  K43 + SRn ×  K35 + SRn+1 ×  K33 +	
	SRn+2 ×  K31 + SRn+3 ×  K29 + SR2n ×  K21 +	
	SR2n+1 ×  K19 + SR2n+2 ×  K17 + SR2n+3 ×  K15	
	SR3n ×  K7 + SR3n+1 ×  K5 + SR3n+2 ×  K3 + SR3n+3 ×  K1	
The coefficients of each row of the kernel are appended and numbered in ascending order. Example. Kn×n is K1, K2, K2, …Kn2	

Supplemental Information

Supplemental Information 1 Matlab code and vivado code

Click here for additional data file.

Supplemental Information 2 FPGA based Real time Deconvolution Accelerator

Click here for additional data file.

Supplemental Information 3 Upsampling raw data and result

The histogram of the images obtained from Matlab and FPGA for realistic comparison

Click here for additional data file.

Appendix

Additional Information and Declarations

Competing Interests

Author Contributions

Data Availability

1 The MATLAB code is provided where we compare the upsampled outputs obtained from Eqs. (4) to (7) with the MATLAB built-in command. Figshare DataPort DOI: 10.6084/m9.figshare.19387118.

2 Vivado project file for the 4 × 4 to 8 × 8 has been uploaded at Figshare Dataport DOI: 10.6084/m9.figshare.13668644.

The authors declare there are no competing interests.

Alex Noel Joseph Raj conceived and designed the experiments, performed the experiments, analyzed the data, performed the computation work, prepared figures and/or tables, authored or reviewed drafts of the paper, and approved the final draft.

Lianhong Cai conceived and designed the experiments, performed the experiments, analyzed the data, performed the computation work, prepared figures and/or tables, authored or reviewed drafts of the paper, and approved the final draft.

Wei Li performed the experiments, analyzed the data, authored or reviewed drafts of the paper, and approved the final draft.

Zhemin Zhuang analyzed the data, authored or reviewed drafts of the paper, and approved the final draft.

Tardi Tjahjadi analyzed the data, authored or reviewed drafts of the paper, and approved the final draft.

The following information was supplied regarding data availability:

The data and VIVADO files are available at Figshare and in the Supplemental Files:

Joseph Raj, Alex Noel (2022): project_4_to_8.zip. figshare. Software. https://doi.org/10.6084/m9.figshare.13668644.v2

Joseph Raj, Alex Noel (2022): matlab code. figshare. Software. https://doi.org/10.6084/m9.figshare.19387118.v2.

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
