# Peer review of "FPGA-based systolic deconvolution architecture for upsampling"

_PeerJ Computer Science, doi:10.7717/peerj-cs.973_

## Round 0.1 · original submission · Major Revisions

Revise as per reviewers' comments.

Reviewer 1 ·

Basic reporting

There seems to be an issue with citation through out the text, for example in line 59 "Lu et al. Lu et al. (2017)" should be "Lu et al. (2017)". This repeated referencing should be an easy fix by fixing the format file.

In line 128 " (i) Interpolation techniques" and line 129 "(iii) Transpose Convolution need references." citations are needed.

Experimental design

No comment.

Validity of the findings

No comment.

Reviewer 2 ·

Basic reporting

This paper presents a deconvolution accelerator that upsample n×n input to 2n×2n output by convolving with a k ×k kernel. The proposed architecture does not insert and pad zeros and thus removes
the redundant computations to achieve high resource efficiency with reduced number of multipliers
and adders.
The paper is well-written and easy to follow.
Adequate literature reviews has been discussed and background clearly explained.
The paper structured well along with formatted plots and figures.
The result section presents a good proof of the established hypothesis.

Experimental design

The article has articulated well around the defined research question.
The results are reproducible and well presented.

Validity of the findings

The results have been evaluated against benchmarks, the dataset has been explained and conclusions are stated clearly along with highlighted contributions.

Reviewer 3 ·

Basic reporting

Abstract:
* Please clearly specify the precision used for operations like multiply/add etc in the proposed architecture.
* Please provide SNR reported by prior works as a point of reference. Further, please specify how the SNR translate to increased accuracy compared to prior work?
* Please use IEEE double precision instead of MATLAB double precision, if applicable.
* Compared to prior work, is the performance higher or lower?

Introduction:
* Line 26-29: It is unclear why the citation [Daoud and Bayoumi, 2019] is used here.
* Line 45-98: Please fix the citations. There are two authors cited for the same work. Example Line 48-49.
* Line 59: Please clarify what is segmentation cost and how it is different from throughput. Is the segmentation cost the energy consumption or latency or something else?
* Line 73: Yazdanbaksh et al has been cited but not explained.
* Line 71: Only suitable for small kernels — why? Is it due to arithmetic/compute complexity or memory consumption.
* Further, the metric used by the paper for evaluating the error

The reported 3.6 GOps at 200 MHz would translate to just 18 Ops/cycle. Does the proposed architecture fully utilize the resources in the FPGA?

Please comment on how the proposed architecture would support neural networks where multiple different deconvolutions layers with different kernel sizes and input/output sizes are used.

Experimental design

The experiments focus on deconvolution layers of fixed sizes (for example, 256x256 output and 3x3 kernel). It is unclear if the evaluated sizes for inputs/output/kernels/strides are employed in real-world workloads like UNets.

The numbers in Table 4 and Table 5 are generated from three sets of images - randomized, camera man, and natural. In contrast, prior works report end-to-end neural network accuracy degradation on widely used neural networks and datasets. As such, it is difficult to fully understand how good/bad the reported SNR is.
To rectify this, please evaluate the impact on accuracy when using the proposed architecture for deconvolution/upsampling layers and a traditional CPU/GPU for the rest in widely used neural networks (like UNET) and datasets.

Please clearly report the utilization of resources for the proposed architecture. The reported 3.6 GOps and 0.135 GOps/DSP would translate to just 27 DSPs. In this case, the proposed architecture does not fully utilize all the DSPs available on the FPGA zc7020.

Validity of the findings

The comparison in Table 10 is misleading since the proposed architecture only supports deconvolution layers of a neural network, while the cited prior works (Di et al, 2020) report end-to-end performance and energy/power efficiency. Please clarify in the table, perhaps by adding a row stating whether the evaluated architectures support layers other than deconvolution.

---

## Round 0.2 · accepted · Accept

The authors have incorporated all the comments. It can be accepted now.